# Behavior and Mechanism of Cesium Biosorption from Aqueous Solution by Living *Synechococcus* PCC7002

**DOI:** 10.3390/microorganisms8040491

**Published:** 2020-03-30

**Authors:** Runlan Yu, Hongsheng Chai, Zhaojing Yu, Xueling Wu, Yuandong Liu, Li Shen, Jiaokun Li, Jun Ye, Danchan Liu, Tao Ma, Fengzheng Gao, Weimin Zeng

**Affiliations:** 1School of Minerals Processing and Bioengineering, Central South University, Changsha 410083, China; yurunlan0715@126.com (R.Y.); 175611014@csu.edu.cn (H.C.); yzj316609308@126.com (Z.Y.); wxlcsu@csu.edu.cn (X.W.); yuandong_liu@csu.edu.cn (Y.L.); lishen@csu.edu.cn (L.S.); lijiaokun@csu.edu.cn (J.L.); 2Hunan Qingzhiyuan Environmental Protection Technology Co, Ltd., Changsha 410000, China; huizhang007@163.com (J.Y.); LDC021411208@126.com (D.L.); 13755167937@139.com (T.M.); 3College of Food Science and Engineering, Ocean University of China, Qingdao 266003, China; gaofengzheng@outlook.com

**Keywords:** Biosorption, *Synechococcus* PCC7002, Cs, EPS

## Abstract

Many efforts have focused on the adsorption of metals from contaminated water by microbes. *Synechococcus* PCC7002, a major marine cyanobacteria, is widely applied to remove metals from the ocean’s photic zone. However, its ability to adsorb cesium (Cs) nuclides has received little attention. In this study, the biosorption behavior of Cs(I) from ultrapure distilled water by living *Synechococcus* PCC7002 was investigated based on kinetic and isotherm studies, and the biosorption mechanism was characterized by Fourier-transform infrared spectroscopy, Raman spectroscopy, scanning electron microscopy, transmission electron microscopy, energy-dispersive X-ray spectrometry, and three-dimensional excitation emission matrix fluorescence spectroscopy. *Synechococcus* PCC7002 showed extremely high tolerance to Cs ions and its minimal inhibitory concentration was 8.6 g/L. Extracellular polymeric substances (EPS) in *Synechococcus* PCC7002 played a vital role in this tolerance. The biosorption of Cs by *Synechococcus* PCC7002 conformed to a Freundlich-type isotherm model and pseudo-second-order kinetics. The binding of Cs(I) was primarily attributed to the extracellular proteins in EPS, with the amino, hydroxyl, and phosphate groups on the cell walls contributing to Cs adsorption. The biosorption of Cs involved two mechanisms: Passive adsorption on the cell surface at low Cs concentrations and active intracellular adsorption at high Cs concentrations. The results demonstrate that the behavior and mechanism of Cs adsorption by *Synechococcus* PCC7002 differ based on the Cs ions concentration.

## 1. Introduction

The risks posed by radionuclides in water have attracted considerable attention since the Fukushima Daiichi Nuclear Power Plant in Japan was damaged by an earthquake [1]. Radioactive fallout from nuclear plants and the nuclear industry can cause great damage to human health upon entering the environment [2,3]. Radiocesium (^137^Cs) is a commonly studied fission product that presents serious concern because of its long half-life of 30 years and ready dissolution in water [4]. Previous studies have applied solvent extraction, ion exchange, and chemical precipitation to remove radionuclides from contaminated wastewater. However, these methods are regarded as inefficient and costly for the removal of Cs [5]. To date, many efforts have focused on identifying materials (e.g., alum and ferric chloride) for removing Cs from aquatic environments [6].

Many efforts have focused on the development of applicable processes and materials for the removal of radioactive fallout [7]. However, few works have focused on the behavior and mechanism of Cs adsorption from aquatic environments. Understanding cyanobacterial metal adsorption and detoxification can promote the utilization of biosorbents and provide a theoretical basis for industrial applications. Many studies have evaluated the biosorption mechanisms of metals in aqueous solution. The adsorption process can be primarily divided into active adsorption and passive adsorption. Active adsorption is further divided into surface complexation, electrostatic adsorption, inorganic microprecipitation, ion exchange, and redox [8,9,10], while passive adsorption is further divided into inorganic microprecipitation and enzymatic mechanisms [11]. For example, when using active brewer’s yeast to adsorb Cd^2+^, Cd^2+^ precipitated as a phosphate in the intracellular vacuoles instead of on the cell walls [12]. Thus, the mechanism of biosorption can be very complex, and multiple mechanisms often occur simultaneously.

Biological materials and biomass have been proposed as effective and economical alternatives to nonbiological materials for the elimination of heavy metals and radioactive pollutants from contaminated water [2,13]. In particular, the removal of Cs using microorganisms, including microalgae, has attracted considerable attention because of the good adsorption ability, low cost, and environmental friendliness of microorganisms [14]. Microalgae such as *Haematococcus pluvialis* and *Chlorella vulgaris* were reported to remove almost 95% and 90% of soluble ^137^Cs, respectively [15]. The abilities of natural algae and chemically modified algae to adsorb Cs were also investigated, revealing the great potential of algae for removing Cs from wastewater with low contents of radionuclides [2].

As the first tool of microorganisms to adapt to adverse environments, extracellular polymeric substances (EPS) can prevent heavy metal ions from entering the cells of the microorganism [16]. Considering their minimal nutritional requirements, easy cultivation and rapid metal removal kinetics, EPS-producing cyanobacteria can be an advantageous choice than other microorganisms for treatment of metal wastewater [17]. Cyanobacteria, or blue-green algae, are common microorganisms with the capacity of producing a great deal of EPS [18]. The cyanobacterial EPS have plenty of negatively charged groups, including different monosaccharides, ester-linked acetyl groups and sulfate groups. Hence, the cyanobacterial EPS show greater complexity than those secreted by other bacteria [19]. Extensive research on the interactions between cyanobacterial EPS and metal ions has been carried out in recent years. However, the role of EPS in mediating heavy metal toxicity has not been extensively discussed. Generally speaking, EPS are secreted by cyanobacteria to help the cyanobacteria adapt to adverse environments, and metal stimulation can increase the production of EPS [20]. According previous studies, the interaction between cyanobacteria and metal ions including exterior precipitation, surface adsorption, intracellular accumulation, and ion exchange [21].

*Synechococcus* PCC7002 is a model marine cyanobacteria, which contributes significantly to metal treatment in the ocean [22]. Nevertheless, there are few reports relating with the metal ions adsorption behavior and mechanism of marine cyanobacteria species. By report, Cd biosorption by *Synechococcus* PCC 7002 revealed that carboxyl groups attributed greatly to adsorption, and the maximum adsorption amount of Cd is 7.05 × 10^−5^ M/g dry biomass in solution. It means that *Synechococcus* PCC7002 show good adsorption capacity of Cd, and *Synechococcus* PCC7002 might be suitable for metal removal [22].

As far as know, Biosorption of Cs nuclides by living *Synechococcus* PCC7002 has not been reported on any literatures in publicly available repositories. Hence, the objective of this work was to investigate the adsorption properties and mechanism of Cs ions by *Synechococcus* PCC7002 to further understand the biosorption behavior and the interactions between marine microorganisms and Cs ions.

## 2. Materials and Methods

### 2.1. Cyanobacteria and Growth Conditions

The cyanobacteria strain *Synechococcus* PCC7002 was kindly provided by the College of Food Science and Engineering, Ocean University of China. It was cultured in 250-mL conical flasks under strong light intensity (100 μmol photons∙m^−2^∙s^−1^) at 32 °C in a light shaker (Changzhou jintan, TS-211GZ, china). All cyanobacteria were cultivated in medium A, as previously reported [23,24]. The living cyanobacteria were harvested by centrifugation at 10,000 rpm for 10 min. No other chemical or physical processing was applied before the biosorption study. The minimal inhibitory concentration (MIC) computation method referred to Ball et al. to determine its capacity of Cs tolerance [25].

### 2.2. Cs Solution

A 1000 mg/L stock solution of Cs was prepared by dissolving analytical grade CsCl in ultrapure distilled water. A series of test solutions was prepared by appropriately diluting the stock solution with nitric acid (10%). All glassware for biosorption experiments was washed with ultrapure water to prevent contamination. All chemicals used in this research were purchased from Sinopharm Chemical Reagent Co. Ltd.

### 2.3. Effect of Cs(I) on Synechococcus PCC7002 Growth

The cyanobacteria strain *Synechococcus* PCC7002 was cultured in medium A supplemented with Cs ions (0–9 g/L) at 32 °C while shaking at 150 rpm. Cells were counted by measuring the optical density of each culture at 750 nm (OD_750_) once per day using an ultraviolet spectrophotometer (Shimadzu, UV-2550, Japan). The experiments were carried out in triplicate, and the mean values were calculated from the data.

### 2.4. EPS Extraction and Chemical Analysis

EPS was extracted using a modified heating procedure that was shown to have low cell lysis and high extraction efficiency [26]. All EPS fractions were filtered through 0.45-μm cellulose acetate membranes to obtain pure EPS. After purification, the EPS solution was immediately lyophilized and stored at 4 °C until use. The contents of polysaccharides, proteins, and nucleic acids in the EPS samples were determined using the phenol–sulfuric acid, BCA Kit (Beyotime, P0012S, china), and NanoDrop methods, respectively [27].

### 2.5. Adsorption Experiments

All tests were conducted by adding biomass to 50 mL of Cs solution under stirring at 150 rpm. The wet cells were collected by centrifugation at 8000 rpm for 10 min, and then suspended in ultrapure water. The total volume for adsorption experiments includes 30 mL cyanobacteria suspensions and 20 mL Cs stock of different concentrations. The final metal concentrations range from 0 to 1 g/L in this mixture. The mixture was centrifuged at 10,000 rpm for 10 min. After appropriate dilution with nitric acid, the residual concentration of Cs^+^ in the supernatant was measured by inductively coupled plasma optical emission spectrometry (PerkinElmer, Optima 5300 DV, Waltham, USA).

To determine the optimum pH value for Cs biosorption by *Synechococcus* PCC7002, the solution pH was adjusted in the range of 1–7 using 0.1 mol/L HCl or NaOH.

The amount of adsorbed Cs^+^ was determined as
(1)$$q_{e}=\frac{\left(c_{0}-c_{t}\right)V}{M}$$
where *C*_0_ (mg/L) is the initial concentration of Cs ions, *C_t_* (mg/L) is the concentration of Cs ions at a given time *t*, *V* (L) is the volume of suspension, and *M* (g) is the dry weight of biomass.

### 2.6. Surface Analysis

The surface functional groups of *Synechococcus* PCC7002 before and after Cs biosorption (at 500 mg/L) were evaluated by Fourier-transform infrared (FTIR) spectroscopy (Thermo, Nicolet NEXUS 670, Waltham, USA) and Raman spectroscopy (HORIBA, LabRAM HR Evolution, France). Firstly, the wet cell samples were dried in a vacuum drying oven, and one part of dried samples was directly for Raman spectroscopy analysis. Then, 1 mg dried cells were mixed with 100 mg of spectrochemical pure potassium bromide (KBr) for FTIR analysis.

The three-dimensional excitation emission matrix (3D-EEM) fluorescence spectra of the collected EPS samples were measured using a Hitachi F-7000 fluorescence spectrophotometer (Hitachi High Technologies, Japan) with a 700-V xenon lamp at a scan speed of 1200 nm/min. The slit of excitation and emission was set to 10 nm. The EEM spectra were collected in 5-nm increments over the excitation range of 220–400 nm and the emission range of 250–550 nm.

### 2.7. Transmission Electron Microscopy (TEM), Scanning Electron Microscopy (SEM), and Energy-Dispersive X-ray Spectroscopy (EDX) Analyses

After Cs biosorption, the biomass suspension was washed three times with ultrapure water. Ultrathin sections of biomass were obtained using an ultramicrotome and fixed on copper grids. The biomass ultrastructure was then observed by TEM (JEOL, JEM-2100F, Japan) at 200 kV. The surface morphologies and structures of *Synechococcus* PCC7002 at different states in the adsorption process were evaluated by field-emission SEM (FEI, NovaNano SEM 230, Hillsboro, USA) after gold plating. SEM can be used to directly observe changes in the surface microstructures of biosorbents upon chemical modification or biosorption [28]. The presence of Cs ions and the chemical composition of *Synechococcus* PCC7002 before and after metal uptake were evaluated by EDX (OXFORD, Aztec Energy X-max 80, England).

### 2.8. Statistical Analysis

The experimental results were performed in triplicate and the data were expressed as the means standard error. One-way analysis of ANOVA in SPSS version 21.0 (SPSS Inc., Chicago, IL, USA) was carried out to confirm the statistical variability and the data were considered significant when *p* value less than 0.05.

## 3. Results

### 3.1. Impacts of Cs(I) on Cyanobacterial Growth

The growth curve of *Synechococcus* PCC7002 was recorded to explore the tolerance of the cyanobacterium to Cs ions. As shown in Figure 1, the growth rate of *Synechococcus* PCC7002 gradually declined as the concentration of Cs ions increased. When the concentration of Cs ions reached 9 g/L, cell growth was completely inhibited. According to metal tolerance results, the minimal inhibitory concentration was 8.6 g/L for *Synechococcus* PCC7002 to Cs.

### 3.2. Effects of Cs(I) on EPS Production and Composition

Figure 2 shows the levels and compositions of EPS produced by *Synechococcus* PCC7002 in the presence of different concentrations of Cs ions. In the absence of Cs ions, the total content of EPS was 72 mg/g dry cell, and the secretion of extracellular proteins (40 mg/g) was higher than that of exopolysaccharides (35 mg/g). When the cyanobacteria was exposed to Cs ions, total EPS first increased and then decreased with increasing Cs concentrations. Total EPS production reached the highest level (110 mg/g dry cell) when the Cs ion concentration was 2 g/L. Similar trends were observed in the contents of extracellular proteins and exopolysaccharides; the protein/polysaccharide ratio ranged from 1.52 to 1.31 as the Cs ion concentration increased from 1 to 4 g/L.

### 3.3. Adsorption Kinetics of Cs(I) on Synechococcus PCC7002

As shown in Appendix A, both Cs adsorption speed and amount increased as the initial Cs ions concentration increased. Adsorption speed was high in the first 10 min and reached Equilibrium after approximately 30 min, at which point the adsorption rate was lower. Lagergren’s equation has been used in past studies to investigate the kinetics of metal ion adsorption onto biomass in wastewater. The linear, first-order Lagergren’s equation is given as [29]:
(2)$$\ln\left(q_{e}-q_{t}\right)={\mathrm{lnq}}_{e}-k_{1}$$
while the linear, pseudo-second-order kinetic equation is represented as:(3)$$\frac{t}{q}=\frac{1}{K_{2}q_{e}{}^{2}}+\frac{t}{q_{e}}$$
where *q_e_* and *q_t_* are the sorbate uptakes at equilibrium (mg/g) and at time *t* (mg/g), respectively; and *k_1_* and *k_2_* are the kinetic rate constants for the pseudo-first-order equation (min^−1^) and the pseudo-second-order equation (g/mg/min), respectively. Figure 3 shows the linear curves of Cs(I) adsorption onto *Synechococcus* PCC7002 for the above two kinetic models; the corresponding kinetic parameters are shown in Table 1.

As shown in Table 1, the correlation coefficient (*R^2^*) of the pseudo-second-order model was superior to that of the first-order model. The pseudo-second-order model provided a perfect fit to adsorption when the initial Cs ions concentration ranged from 300 to 700 mg/L. Hence, the pseudo-second-order model was more suitable for describing the biosorption kinetics of Cs ions by *Synechococcus* PCC7002 than the first-order model. In the Lagergren second-order model, the estimated equilibrium uptake (q_e_ values) were 25.6 mg/g and 56.27 mg/g at 0.3 g/L and 0.7 g/L Cs ion concentration, respectively.

Langmuir-type sorption occurs when the intermolecular forces decrease quickly with distance; that is, the surface of the biosorbent is covered by adsorbate, which forms a layer on the biomass. The linear Langmuir isotherm is given by [30]:(4)$$\frac{c_{e}}{q_{e}}=\frac{c_{e}}{q_{m}}+\frac{1}{q_{m}b}$$
where *C_e_* is the concentration of metal ion at equilibrium (mg/L), *q_e_* is the amount of adsorbed metal (mg/g), *q_m_* is the amount of Cs adsorbed at equilibrium (mg/g), and *b* is the Langmuir constant (mg/L). In contrast, under the Freundlich model, the surface of the biosorbent is heterogeneous. The Freundlich model is given by:(5)$$\ln q_{e}=lnK_{F}+\frac{1}{n}lnC_{e}$$
where *K_F_* and *n* are Freundlich constants that are related to the adsorption capacity and the sorption intensity, respectively.

To clarify the sorption behavior and understand the surface characteristics of the cyanobacteria, the adsorption isotherm was investigated at different temperatures based on the Langmuir and Freundlich models. The Cs adsorption isotherms are shown in Figure 4, and the experimentally determined isotherm parameters are shown in Table 2. The *R*^2^ value of the Freundlich adsorption isotherm was better than that of the Langmuir adsorption isotherm. Moreover, among the two isotherm models, the experimental Cs adsorption data were better fit by the Freundlich isotherm, which accurately described the biosorption of Cs ions onto *Synechococcus* PCC7002 in the temperature range of 20 °C to 40 °C. When the temperature increased from 20 to 40 °C, an increase in q_max_ obtained from fitting the Langmuir isotherm was observed (126.74 mg/g at 20 °C to 146.84 mg/g at 40 °C).

### 3.4. Effects of Cs(I) on the Chemical Functionalities of Synechococcus PCC7002

To identify the effects of Cs(I) adsorption on the cell wall/membrane chemistry, the FTIR and Raman spectra of *Synechococcus* PCC7002 were collected before and after Cs adsorption (Figure 5). The FTIR spectrum of *Synechococcus* PCC7002 before adsorption shows bands at 3298, 2922, 1542, 1456, 1234, and 1061 cm^−1^, corresponding to O–H, CH_2_, N–H, CH_2_, and PO^2−^ groups, respectively. Thus, the FTIR spectrum indicates numerous functional groups on *Synechococcus* PCC7002, including amino, hydroxy, and phosphate groups. After Cs adsorption, a new FTIR peak appeared at 1659 cm^−1^, indicated that the amide I groups of α-helical structures played an important role in Cs adsorption. In addition, a shift in the peak at 1234 cm^−1^ was observed, indicating that PO^2−^ asymmetric stretching was involved in Cs^+^ binding. Finally, the small peak shifts from 1456 to 1451 cm^−1^ and from 3298 to 3295 cm^−1^ indicate that CH_2_ bending, N–H stretching, and O–H stretching were also involved in Cs adsorption.

More structural information about *Synechococcus* PCC7002 before and after the biosorption of Cs was obtained by Raman spectroscopy (Appendix A). After Cs adsorption, changes were observed in the Raman peaks at 984, 1006, 1154, and 1524 cm^−1^, which were tentatively assigned to saturated C–H and C–C of cyclobutene, ring deformation of phenylalanine, C–C stretching, and C–H_2_ deformation, respectively. These bands can likely be attributed to the numerous functional groups in the saccharides and amino acid side chains of proteins. After adsorption, the intensities of the peaks at 1006, 1154, and 1524 cm^−1^ decreased, and a peak shift from 984 to 972 cm^−1^ was observed.

### 3.5. Effects of Cs(I) on the Surface Morphology and Composition of Synechococcus PCC7002

The surface morphology and composition of *Synechococcus* PCC7002 were analyzed before and after Cs adsorption by SEM and EDX. As shown in Figure 6, the surface of the biomass clearly became rougher after adsorption. According to EDX analysis, the proportions (W/W) of carbon, oxygen, sodium, phosphorus, chlorine, and potassium before adsorption were 54%, 32%, 5.6%, 5.38%, 1.67%, and 0.14%, respectively (Figure 7a). After Cs adsorption, the proportions of phosphorus and potassium increased to approximately 15.17% and 3.17%, respectively, and two new peaks corresponding to Cs ions emerged (Figure 7b), confirming the presence of Cs ions on the surface of *Synechococcus* PCC7002.

### 3.6. Effects of Cs(I) on EPS

To further investigate the role of EPS in Cs ion adsorption by *Synechococcus* PCC7002, EEM fluorescence spectroscopy was used to assess the content and composition of EPS before and after adsorption (Figure 8). The EEM spectra of EPS clearly showed five fluorescence peaks both before and after Cs adsorption. Peaks A (Ex/Em = 230 nm/340 nm) and C (Ex/Em = 280 nm/345–350 nm) were ascribed to aromatic protein substances (Ex < 250 nm, 330 nm < Em < 380 nm) and tryptophan and protein-like substances in the region of soluble microbial by product-like substances (Ex > 250 nm, Em < 380 nm), respectively. Peaks B (Ex/Em = 275 nm/450 nm) and E (Ex/Em = 350 nm/450 nm) were both ascribed to humic acid-like substances (Ex > 250 nm, Em > 380 nm) [31].

The intensities of peaks A and C increased after Cs adsorption, suggesting that some biochemical reactions occurred during the adsorption process. Therefore, aromatic protein substances and tryptophan and protein-like substances were produced in EPS when Cs ions were loaded. However, the intensities of peaks B and E decreased under Cs stress, suggesting that the structure of humic acid in EPS was destroyed by exposure to high concentrations of Cs.

### 3.7. TEM/EDX Analysis of the Localization of Cs(I) in Synechococcus PCC7002 During Adsorption

The TEM analysis of ultrathin sections of cyanobacteria can provide information on the presence, spatial distribution, and morphology of Cs complexes. As shown in Figure 9, after adsorption, both extracellular and intracellular Cs ions were detected in *Synechococcus* PCC7002. When the Cs ions concentration was 0.5 g/L, Cs deposited solely on the cell walls, while significant Cs precipitation within cells was not observed. However, the TEM micrographs of *Synechococcus* PCC7002 exposed to Cs ions at 5 g/L show Cs precipitation on the cell wall and in the periplasmic space; the precipitated Cs mainly accumulated intracellularly, forming large Cs clusters inside the cyanobacteria cells. Thus, Cs ions entered into *Synechococcus* PCC7002 and was not merely fixed on the cell surface.

The EDX peak of C in Figure 9f indicated a C content (W/W) of 1.5%. The proportions (W/W) of Cs detected in biomass before adsorption (Figure 9d) and in biomass exposed to Cs ions at 0.5 g/L (Figure 9e) were 0% and 0.4%, respectively. Thus, the EDX results suggest that Cs ions entered into the cytoplasm during the adsorption process. The presence of copper and lead in the samples can be attributed to the fixing of the samples on copper grids and the use of lead citrate as a staining solution, respectively.

## 4. Discussion

In this study, *Synechococcus* PCC7002 showed good performance for Cs removal, and we investigated the mechanism of the biosorption process. First, we evaluated the growth and EPS production of *Synechococcus* PCC7002 under different concentrations of Cs ions. Cs ions did harm the growth of *Synechococcus* PCC7002. Prior studies found that molecular-level changes (e.g., polypeptide chain misfolding) are the main pathway through which heavy metals cause injury to microorganisms [32]. These changes can result in the production of active oxygen and enzyme inactivation, leading to microorganism death [33]. To reduce the hazards of heavy metals to cyanobacteria, generally speaking, EPS first binds to the metal ions; thus, EPS plays a significant role in the resistance of microorganisms to heavy metals [34]. Several reports have shown that EPS with abundant electron-donating functional groups can be used to precipitate heavy metals, and the negatively charged cell walls of microorganisms facilitate the binding of cationic metals [35,36]. Nevertheless, at high metal concentrations, the metabolic activity of *Synechococcus* PCC7002 decreases, and the secretion of EPS is reduced [37]. Furthermore, some studies have suggested that intracellular absorption rather than EPS sedimentation may increase microbial tolerance to Cs(I) under high concentrations of metal ions [38]. In this study, both the protein and polysaccharide contents in EPS showed similar trends with Cs ion concentration, although the protein content showed greater variation. These findings suggest that under Cs stress, *Synechococcus* PCC7002 secretes extracellular proteins that protect the cells by binding and precipitating Cs ions.

To understand the Cs biosorption process, we must understand two basic elements: the adsorption kinetics and adsorption equilibrium. Adsorption kinetics reflect the chemical properties over time and are particularly concerned with the adsorption rate. Understanding adsorption behavior can provide insights into various practical applications, including catalysis, corrosion, and the removal of metal ions from solution [39]. The results of this study demonstrate that chemisorption might be the rate-limiting step rather than mass transfer in solution. The Langmuir and Freundlich models have been used by many researchers to describe the adsorption process [40,41]. In this study, *K_F_* increased with increasing temperature, indicating that the adsorption capacity of *Synechococcus* PCC7002 for Cs increased with temperature. At all studied temperatures, *n* was not less than unity, suggesting a favorable biosorption processes [42]. These results indicate that the adsorption process might be heterogeneous, suggesting different adsorbate–adsorbent affinities at different active centers of the adsorbent, and that the process can be well explained by monolayer adsorption. However, the Freundlich isotherm model is empirical and fails at high concentrations of adsorbate near the maximum adsorbent capacity [43]. Hence, the Freundlich model cannot describe the maximum adsorption capacity or be used to predict the adsorption behavior at concentrations outside of the relevant range.

The adsorption mechanism was characterized in this study by using FTIR spectroscopy, Raman spectroscopy, SEM, TEM, EDX, and 3D-EEM analyses. While both Raman and FTIR spectroscopies have advantages and disadvantages, the chemical structural data provided by these methods are complementary [44]. The vibrational bands typical of proteins, polysaccharides, and nucleic acids can be detected by Raman spectroscopy, and micro-Raman spectroscopy can be used to monitor the appearance and components of EPS in microorganisms [45,46]. In this study, the combined FTIR and Raman spectroscopic results indicated that the amide, hydroxyl, carboxyl, and phosphate groups on the cell walls along with the proteins and polysaccharides in EPS played a role in Cs ion adsorption.

The adsorption of metal ions is known to be promoted by the production of EPS, which facilitates the deposition of metal ions on the biomass surface [47]. This effect is attributed to the abundant functional groups of EPS, which include hydroxy, carboxyl, phosphoric amine, and hydroxyl groups and promote the formation of metal ion clusters [48]. Furthermore, the EPS of cyanobacteria show stronger binding capacity for metal ions than EPS from other bacteria due to the complex EPS composition and abundance of negatively charged functional groups (e.g., ester-linked acetyl groups and peptide moieties and sulfate groups [26]). Recently, EEM fluorescence spectroscopy has been applied to investigate the structural and compositional variations in EPS before and after loading with heavy metal ions, including zinc and copper [49]. In this study, the fluorescence peaks of C in EPS shifted toward longer wavelengths (redshift) upon Cs adsorption (Figure 8), which was associated with increases in the content of carbonyl, hydroxyl, alkoxyl, amino, and carboxyl-containing groups [50]. Consequently, we can conclude that the extracellular proteins played a vital role in binding Cs ions. Interestingly, in this study, the polysaccharides were not the main components of EPS under Cs stress; this finding is in consistent with an earlier report on the accumulation of Cs in *Nocardiopsis* sp. 13H [51]. Therefore, the production of EPS clearly differs among different strains under Cs stress. Cs ions can stimulate *Synechococcus* PCC7002 to produce specific extracellular proteins that protect the cells by binding with metal ions.

Many studies have confirmed the irregular shapes and sizes of biomass surface, which have been suggested to facilitate metal ion adsorption [52,53]. Microprecipitation was recognized as an important mechanism of heavy metal removal form aqueous solution via biosorption [54]. However, in this study, the cell surface remained clean without clear metal deposition after Cs(I) biosorption. Based on the SEM/EDX results, ion exchange rather than microprecipitation was the primary mechanism of Cs(I) removal from aqueous solution in this study. The potassium and phosphorus ions released from cell surface after Cs ions into the solution. Combined with the dependence of Cs adsorption on pH (Appendix A), this suggests that the R–COOH groups on the cell walls are likely sites of ion exchange. Other functional groups present on the cell walls of algae may also participate in adsorption, including R–OSO_3_ groups in fucoidan and carrageenan and amino groups [55]. To determine whether Cs ions were adsorbed on the cell surface or entered into the cell, *Synechococcus* PCC7002 was analyzed by TEM before and after Cs ions adsorption. The results agreed with a previous report on Cs ions uptake by *Rhodosporidium fluviale* strain UA2 [56]. Based on the TEM images, Cs clearly precipitated in various positions, sizes, and shapes. The Cs particles were likely distributed intracellularly or combined with intracellular proteins in the cytoplasm [57]. However, when the concentration of Cs ions increased, EPS secretion decreased, and potassium ion transport systems were activated. Thus, as shown in Figure 9c, Cs ions were able to enter into the cytoplasm [58]. In general, we can conclude that the adsorption of Cs by *Synechococcus* PCC7002 occurred via two mechanisms. At low Cs concentrations, Cs ions adsorbed rapidly on the cells, and EPS were produced to resist the toxicity of Cs. However, at high Cs concentrations, Cs ions entered into the cytoplasm through the cytomembrane [59].

Conventional techniques for the removal of heavy metals, including metal precipitation and chelating agents, have proven ineffective for Cs removal [60]. Unlike the primarily metabolism-independent adsorptive behavior of other heavy metals, the predominantly monovalent Cs is accumulated intracellularly via energy-dependent transport in exchange with potassium ion [60]. Compared to other nuclides, like uranium and strontium, more research finds that intracellular uptake is independent of metabolism. For example, uranium precipitates in cells because of the high extracellular osmotic pressure and complexation with polyphosphate [61,62]. As a result, Cs is considered separately here. Furthermore, among bacteria, cyanobacteria possess the most abundant monovalent cations, and the heavy metal-binding protein metallothionein has been confirmed to exist in the prokaryote *Synechococcus* PCC7002 [63]. Since the ionic radii of potassium and Cs are similar, many low-selectivity potassium ion transport systems on the microbial surface cannot distinguish between these ions. Hence, Cs ions may substitute for potassium ions and cross the cell membrane via metabolism-dependent transport processes [64]. To reduce metal toxicity in cells, in addition to vacuole and intracellular deposition, metal-binding proteins can remove heavy metals to enhance microbial tolerance to metals [65].

We could not get an exact maximum capacity (q_max_ value), because the experimental data did not fit to Langmuir model. When Cs ion concentration was 0.5 g/L, the adsorption amount was 30 mg/g at pH 5. Batch experiments of biosorption were conducted to determine the cesium binding ability of native biomass, and the cesium sorption performances of various types of sorbents were compared using the maximum capacities (q max values) obtained from the Langmuir isotherm. The q_max_ value of *P. australis* and *M. somalensis* in this study was 16.2 and 21.9 mg/g, respectively [2]. The adsorption capacity of pretreated arca shell biomass was 3.93 mg/g of Cs, under the optimized condition of initial concentration, pH, biomass dose and contact time [66]. Based on these results, our study showed a good adsorption of Cs. Hence, *Synechococcus* PCC7002 might be suitable for cesium removal.

Microorganisms interact with nuclides through different mechanisms that work together to promote nuclide adsorption. Weak binding between Cs ions and ligands has been reported. Thus, the surface adsorption of Cs may occur primarily through ion exchange and electrostatic interactions rather than covalent bonding, as occurs between sulfur-donor ligands and softer metals [14]. It should be noted that this study did not examine molecular-level interactions, and proteins played a vital role in the cyanobacterial response to Cs stress in this study. Hence, the proteomes of *Synechococcus* PCC7002 in the absence and presence of Cs will be compared in future works. In summary, the Cs adsorption mechanism changed obviously with the concentration of Cs ions. This suggests that Cs biosorption is a complex process that is affected by the metal concentration and other factors. Hence, for the practical treatment of Cs-contaminated wastewater, different measures should be implemented to promote Cs removal under different or variable conditions.

## 5. Conclusions

In this study, *Synechococcus* PCC7002 was applied to remove Cs ions from aqueous solution. The equilibrium Cs uptake process was well described by the Freundlich model, indicating heterogeneous adsorption with monolayer binding. The biosorption process was rapid and followed pseudo-second-order kinetics based on chemisorption. The functional groups binding Cs ions on the cell walls of *Synechococcus* PCC7002 were mainly amine, carboxyl, hydroxyl, and phosphate groups, and the extracellular proteins in EPS played a vital role in the binding of Cs ions. The biosorption of Cs ions by *Synechococcus* PCC7002 involved two mechanisms: Passive biosorption and active biosorption. The findings derived from this work provide valuable information about the biosorption behavior and mechanism of Cs(I) on living *Synechococcus* PCC7002. This work is expected to help identify efficient and economical biosorbents for the removal of Cs(I).

## Figures and Tables

**Figure 1 microorganisms-08-00491-f001:**
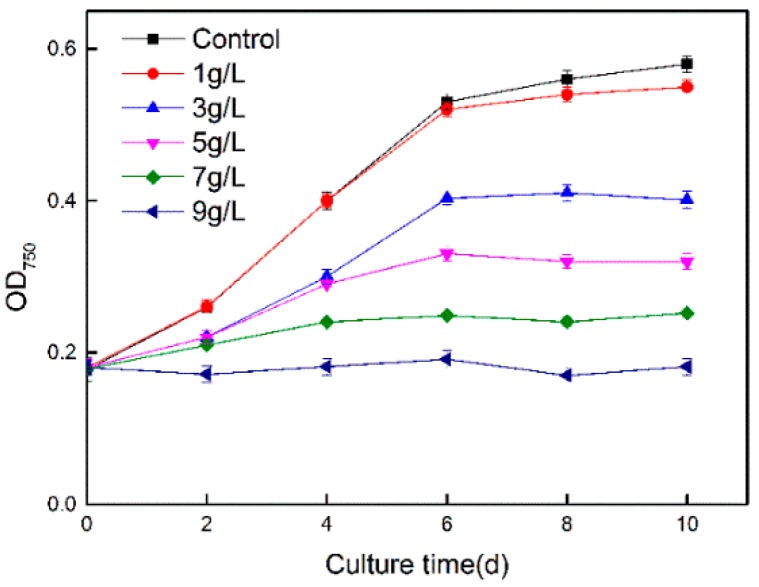
Growth curves of *Synechococcus* PCC7002 under different concentrations of Cs(I).

**Figure 2 microorganisms-08-00491-f002:**
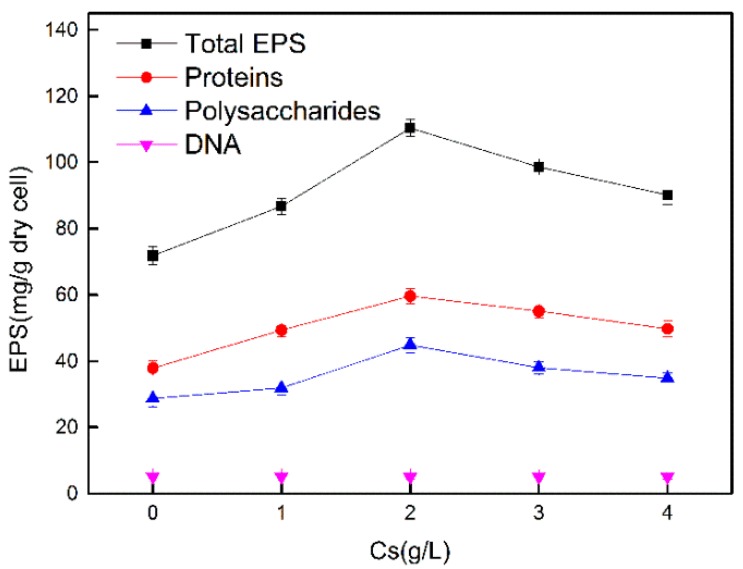
EPS production by *Synechococcus* PCC7002 under different concentrations of Cs(I).

**Figure 3 microorganisms-08-00491-f003:**
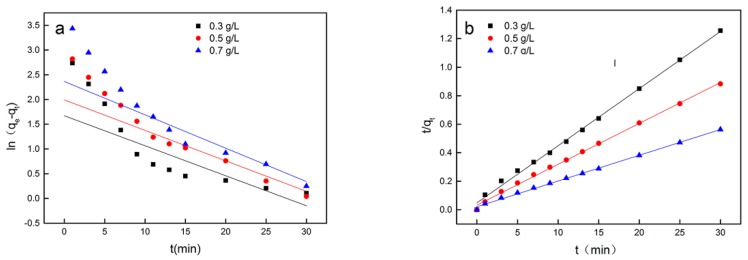
Kinetics of Cs(I) adsorption on *Synechococcus* PCC7002 in aqueous solution fitted using the pseudo-first-order model (**a**) and pseudo-second-order model (**b**).

**Figure 4 microorganisms-08-00491-f004:**
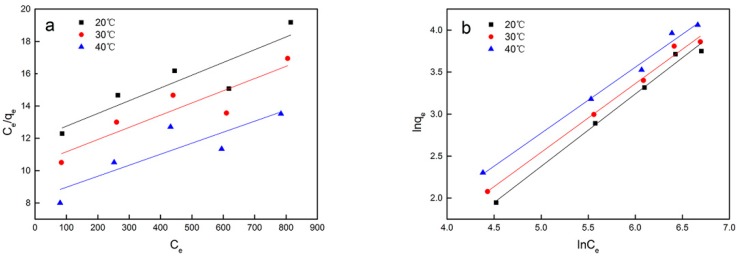
Langmuir isotherms (**a**) and Freundlich isotherms (**b**) of Cs(I) adsorption on *Synechococcus* PCC7002.

**Figure 5 microorganisms-08-00491-f005:**
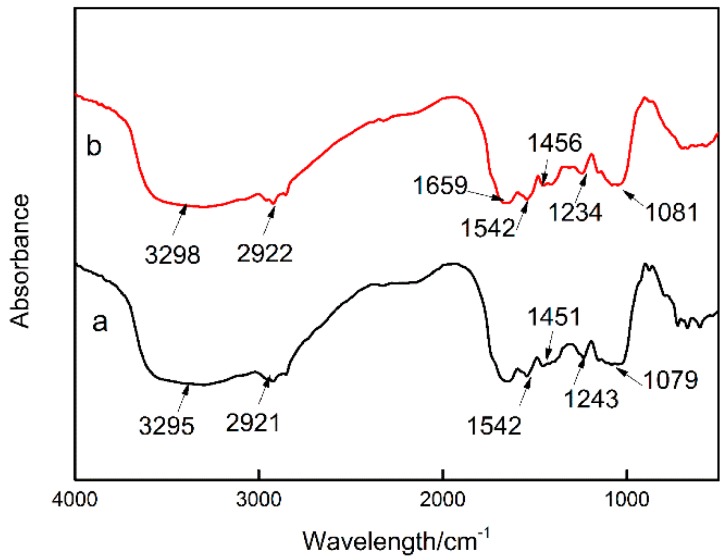
FTIR spectra of *Synechococcus* PCC7002 (**a**) and Cs-loaded *Synechococcus* PCC7002 (**b**).

**Figure 6 microorganisms-08-00491-f006:**
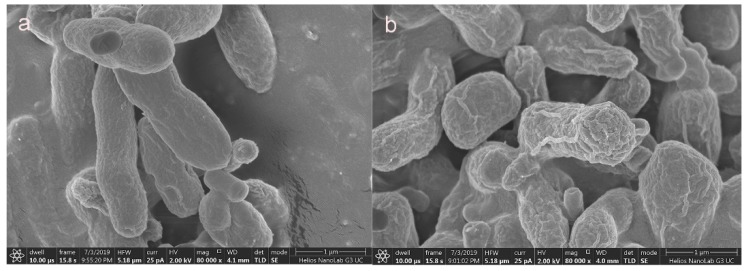
SEM images of *Synechococcus* PCC7002 (**a**) and Cs-loaded *Synechococcus* PCC7002 (**b**). Scale bars: (**a**) 1 μm; (**b**) 1 μm.

**Figure 7 microorganisms-08-00491-f007:**
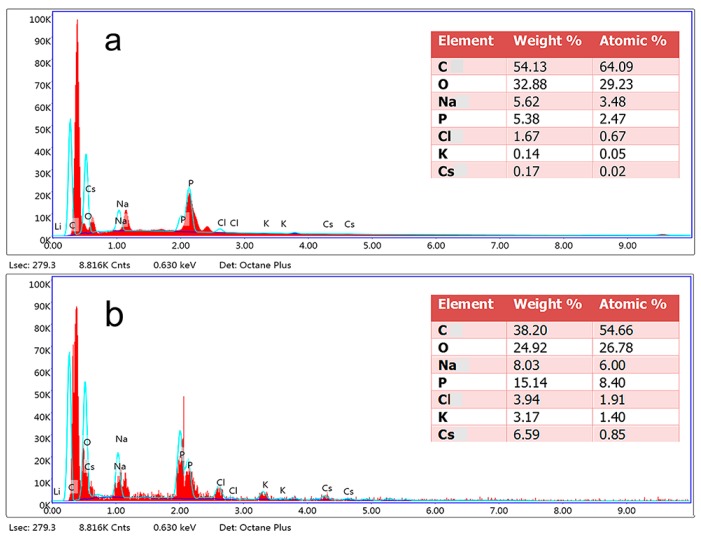
EDX spectra of *Synechococcus* PCC7002 (**a**) and Cs-loaded *Synechococcus* PCC7002 (**b**). The elements analyzed were C, O, Cl, P, Na, K, and Cs.

**Figure 8 microorganisms-08-00491-f008:**
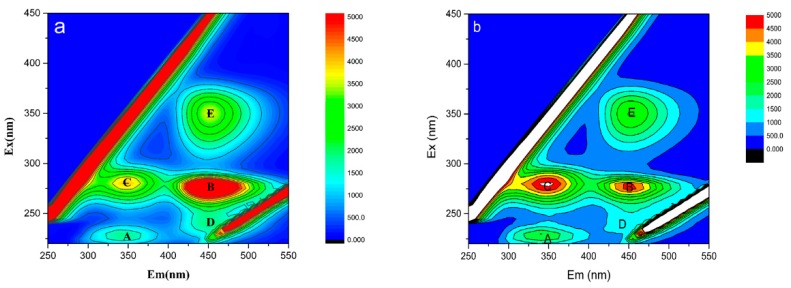
Excitation emission matrix (EEM) fluorescence spectra of extracellular polymeric substances (EPS) from *Synechococcus* PCC7002 (**a**) and Cs-loaded *Synechococcus* PCC7002 (**b**).

**Figure 9 microorganisms-08-00491-f009:**
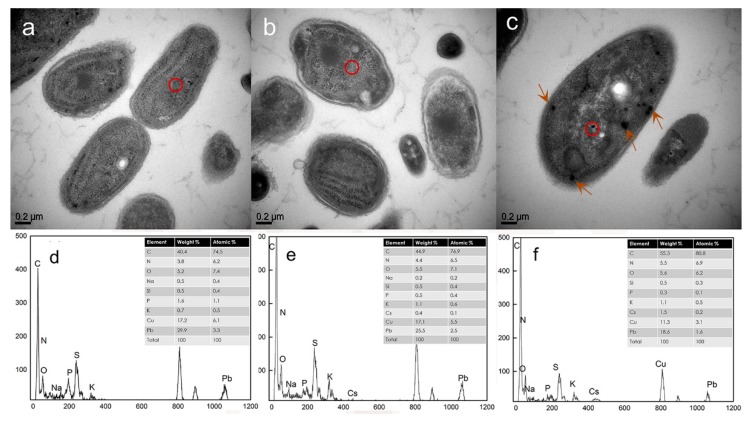
TEM images and corresponding EDX spectra of *Synechococcus* PCC7002 before and after the adsorption of Cs at different concentrations: (**a**,**d**) 0 g/L (untreated biomass), (**b**,**e**) 0.5 g/L, and (**c**,**f**) 5 g/L of Cs ions. The red circles in the TEM images indicate the locations of EDX analysis. The arrows in Figure 9c indicate the precipitates in the cytoplasm, which demonstrate the intracellular uptake of Cs ions.

**Table 1 microorganisms-08-00491-t001:** Kinetic parameters for the biosorption of Cs(I) onto *Synechococcus* PCC7002.

Biomass	Initial Concentration	First-Order Model			Second-Order Model		
	(mg L^−1^)	q_e_	K_1_	R^2^	q_e_	K_2_	R^2^
*Synechococcus*	300	8.49	0.085	0.84	25.4	0.025	0.99
PCC7002	500	12.75	0.091	0.84	35.46	0.019	0.99
	700	20.59	0.102	0.78	56.27	0.011	0.99

**Table 2 microorganisms-08-00491-t002:** Comparison of the Langmuir and Freundlich constants obtained from adsorption isotherms of Cs(I).

Biomass	T/℃	Langmuir Constant			Freundlich Constant		
		B (L mg^−1^)	Q_m_ (mg g^−1^)	R^2^	n	K_F_ (mg g^−1^)	R^2^
*Synechococcus*	20	0.00066	126.74	0.84	1.16	0.145	0.99
PCC7002	30	0.00072	134.05	0.84	1.22	0.214	0.99
	40	0.00082	146.84	0.78	1.27	0.223	0.99

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
