# Peer review of "Behavior and Mechanism of Cesium Biosorption from Aqueous Solution by Living Synechococcus PCC7002"

_microorganisms, 2020, doi:10.3390/microorganisms8040491_

Round 1

Reviewer 1 Report

The MS entitled “Behavior and mechanism of cesium biosorption from aqueous solution by living Synechococcus PCC7002 was focused on the bioremoval of Cs from ultrapure distilled water using living Synechococcus. It has reported that the living Synechococcus can adsorb cesium (Cs) nuclides in two ways such as passive, adsorbs on the cell surface at low Cs concentrations and active, adsorbs intracellularly at high Cs concentrations with instrumental support.

This topic should be of interest to many readers of the ‘Microorganisms Journal’. However, there are minor comments in the manuscript that force me to advise the editor to consider the MS for a minor revision.

Some minor comment:

Abstract:

Page 1, Line 21-22: Synechococcus PCC7002 showed extremely high tolerance to Cs ions- mention the value (concentration).

Line 25-26: This sentence should be followed by Line 22-23.

The authors should mention the source of water as it is real wastewater or pure water.

Line 28: -active intracellular adsorption at high Cs concentrations- adsorption or absorption?

Introduction

This part is not clear and should be reconsidered to follow the flow of metal removal by microbes, algae, and their advantage compared to other microbes and gaps in the study.

Line 19-30: This part focused only on bacteria and their EPS role in metal removal mechanism. Missing the role of algae EPS on metal removal.

Line 28-30: The interlinking of algae into bacterial metal adsorption/mechanism is not clear.

The authors need to explore the advantage of algae instead of choosing bacterial adsorption.

Line 32-33: The sentence-however, few studies have focused on the biosorption mechanism of Cs nuclides by living Synechococcus PCC7002- give the recent references and their observation (removal, adsorption properties, and mechanism) in those studies. Overall, the novelty of this manuscript is missing or needs to explain more clearly.

Materials and methods

Synechococcus is cyanobacteria or algae?

Why had the authors chosen the 32 degrees for the growth of the culture?

Is its optimal growth condition for cyanobacteria?

Does 100 μmol photons∙m−2∙s−1 is strong light intensity?

Page 3, Line 15: Mention the details of biomass added and brief procedure, and Cs concentration range selected for the studies here itself.

Page 3, Line 20: Why the authors have chosen the pH acidic to neutral only for biosorption studies?

Results:

-This section is very confusing. Why the authors have chosen Cs concentration 0-9 g/L for growth estimation and 4g/L for EPS and other parameters, and 0.3 to 0.7 g/L for biosorption studies?

-Missing the data of optimum pH value for Cs biosorption by Synechococcus PCC7002

-Biostatistical values

Change bacteria into cyanobacteria throughout the text.

Discussion

The authors should more concerned to compare the biosorption values and findings with the previous reports to support the novelty of the species selected and this work rather than using any other microorganisms.

Author Response

Response to Reviewer 1 Comments

Point 1: Synechococcus PCC7002 showed extremely high tolerance to Cs ions-mention the value (concentration).

Response 1(page 1, line 22-23. page 3, line 3-4. page 4, line 26-27). We have added the maximum tolerance concentration of Cs by Synechococcus PCC7002 according to the comment.

Point 2: This sentence should be followed by Line 22-23.

Response 2: (page 1, line 24-25). This sentence was modified according to the comment

Point 3: The authors should mention the source of water as it is real wastewater or pure water.

Response 3: (page 1, line 17). We have added the source of water according to the previous comment.

Point 4: active intracellular adsorption at high Cs concentrations-adsorption or absorption?

Response 4: (page 1, line 28). This is active intracellular adsorption.

Point 5: This part is not clear and should be reconsidered to follow the flow of metal removal by microbes, algae, and their advantage compared to other microbes and gaps in the study.

Response 5: (page 1, line 3-4. page 2, line 1-32). To be more clearly and in accordance with the reviewer concerns, we have refreshed the flow of introduction and added cyanobacterial advantage compared to other microbes in the study.

Point 6: This part focused only on bacteria and their EPS role in metal removal mechanism. Missing the role of algae EPS on metal removal.

Response 6: (page 2, line 24-28). We have added the information required as the comment.

Point 7: The interlinking of algae into bacterial metal adsorption/mechanism is not clear.

Response 7: (page 2, line 32-33). To be more clearly and in accordance with the reviewer concerns, we have added the contents of cyanobacterial adsorption/mechanism and removed some irrelevant contents in the revised manuscript.

Point 8: The authors need to explore the advantage of algae instead of choosing bacterial adsorption.

Response 8: (page 2, line 22-25) The contents of the advantage of blue-green algae have been supplemented in the revised manuscript.

Point 9: The sentence-however, few studies have focused on the biosorption mechanism of Cs nuclides by living Synechococcus PCC7002- give the recent references and their observation (removal, adsorption properties, and mechanism) in those studies. Overall, the novelty of this manuscript is missing or needs to explain more clearly.

Response 9: (page 2, line 34-45). We have supplemented the recent adsorption research by Synechococcus PCC7002 in our revised manuscript, and explained more clearly about the novelty of our work.

Point 10: Synechococcus is cyanobacteria or algae?

Response 10: Synechococcus is cyanobacteria.

Point 11: Why had the authors chosen the 32 degrees for the growth of the culture?

Response 11: Because 32℃ is the optimum temperature for the growth of Synechococcus PCC7002.

Reference:

[1] Gao, F., Wu, H., Zeng, M., Huang, M., Feng, G. Overproduction, purification, and characterization of nanosized polyphosphate bodies from Synechococcus sp PCC 7002. Microbial Cell Factories. 2018. 17, (10.1186/s12934-018-0870-6)

Point 12: Is its optimal growth condition for cyanobacteria?

Response 12: The growth condition is the optimal conditions for Synechococcus PCC7002 instead of cyanobacteria. We have replaced “cyanobacteria” by “Synechococcus PCC7002”.

Point 13: Does 100 μmol photons∙m2∙s1 is strong light intensity?

Response 13: 100 μmol photons∙m2∙s1 is a strong light intensity for other cyanobacteria, but do not harm to Synechococcus PCC7002.

Point 14: Mention the details of biomass added and brief procedure, and Cs concentration range selected for the studies here itself.

Response 14: (page 3, line 25-28). We have provided more details of biomass added and brief procedure and the concentration range selected for the studies.

Point 15: Why the authors have chosen the pH acidic to neutral only for biosorption studies?

Response 15: The pH value in real aquatic environment is almost 6-7, and the maximum total adsorption of Cs by P. australis took place at pH 4.

[1] Jalali-Rad, R.; Ghafourian, H.; Asef, Y.; Dalir, S. T.; Sahafipour, M. H.; Gharanjik, B. M. Biosorption of cesium by native and chemically modified biomass of marine algae: introduce the new biosorbents for biotechnology applications. J. Hazard. Mater. 2004. 116, 125-134. (https://doi.org/10.1016/j.jhazmat. 2004.08.022)

Point 16: This section is very confusing. Why the authors have chosen Cs concentration 0-9 g/L for growth estimation and 4g/L for EPS and other parameters, and 0.3 to 0.7 g/L for biosorption studies?

Response 16: To be more clearly and in accordance with the reviewer concerns, we have added a brief description as follows:

We had tested adsorption and tolerances of Synechococcus PCC7002 under a series of Cs concentrations in our previous study. The results showed that the minimal inhibitory concentration (MIC) was 8.6 g/L. However, the adsorption efficiency was extremely low when Cs concentration was 1-9 g/L. Hence, we have chosen Cs concentration 0-9 g/L for growth estimation and 0.3 to 0.7 g/L for biosorption studies. Similarity, total EPS production reached the highest level when the Cs ion concentration was 2 g/L, and decreased when Cs ion concentration more than 4 g/L.

In a word, we have chosen these Cs concentrations for better description of growth estimation, EPS production, and biosorption experiment under different Cs concentrations separately.

Point 17: Missing the data of optimum pH value for Cs biosorption by Synechococcus PCC7002

Response 17: The data have been added in in the supplementary material according to the comment. The pH had a vital influence on the biosorption and accumulation of Cs+, as shown in Figure S2. The amount of biosorbed Cs+ increased sharply as pH increased from 2.0 to 5.0 and then gradually decreased as pH further increased from 5.0 to 7.0. Thus, the optimal pH for Cs biosorption was 5.

Point 18: Biostatistical values

Response 18: (page 5, line27-29. page 6, line 17-18). We have added the Biostatistical values according to the comment.

Point 19: Change bacteria into cyanobacteria throughout the text.

Response 19: We have modified throughout the text according to the comment.

Point 20: The authors should more concerned to compare the biosorption values and findings with the previous reports to support the novelty of the species selected and this work rather than using any other microorganisms

Response 20: (page 12, line 32-40). We have compared the adsorption values of Cs between Synechococcus PCC7002 and other microorganisms, and proved that Synechococcus PCC7002 show good adsorption capacity of Cs. Hence, Synechococcus PCC7002 might be suitable for cesium removal.

Reviewer 2 Report

The work presented by Yu et al. studies the mechanism and behavior of cesium adsorption by a strain of Synechococcus, which is very interesting. To achieve this goal, appropriate experimental procedures and analyses have been employed. I recommend accepting the manuscript after the following revisions.

Specific comments

Introduction

P2, lines 15-17: The example provided, is it active or passive adsorption? Please, clarify this point.

Material and Methods

P2, line 37: Please, replace “Cyanobacteria and cultivation” by “Cyanobacteria and growth conditions”

P2, Line 40: Please, provide the brand of the light shaker.

P3, line 5: Please, clarify whether standard deviations were calculated.

P3, line6: Please, provide the software employed for calculations.

P3, line 12: Please, include the brand of the BCA kit.

P3 line 13: The reference included seems to correspond to sugars only. Please, check it.

P3 lines 24-27: Please, provide more detail on how cells were prepared for FTIR and Raman.

P3, line 37: Please, provide the brand of TEM.

Results

P4, line 7: In this figure legend, please explain what the vertical bars are. Are they standard deviations? Please, clarify it.

P4, line 13: “concentrations”

P4, line 13: Please, replace “Total EPS was maximized” by “Total EPS production reached the highest level”.

P4, line 18: In this figure legend, please explain what the vertical bars are. Are they standard deviations? Please, clarify it.

P5, line 1-33: Please, provide suitable references for these equations.

P5, line 1-33: Please, explain the meaning of the numbers in brackets or remove them. The same for the equation presented in page 3.

P6, Figures 3 and 4: It’s not necessary to denote the type of model employed within the figures as it is indicated in the figure legend.

P7, line 6: PO2- is repeated twice. Symmetric, asymmetric groups? Please, clarify this point.

P7, line 9: The peak at 1659 corresponds to amide I rather than to amide II. Please, clarify this point.

P7, line 16: 1153 and 1516 peaks were not seen in Figure 5

P7, Figure 5: To better follow the results, please represent a different figure for each FTIR and Raman spectra.

P8, Figure 7: Please, indicate the meaning of the values at the axes. Also, please remove the letter K following the element in the legend within the figure.

P8, line12: “by product”

P8, line15: Please, replace “indicating” by “suggesting”.

P8 line 18: “Cs ions were loaded”. Please, remove “in or on Synechococcus PCC7002”.

P9, line 1: Please, replace “indicating” by “suggesting”.

P9, Figure 8: Please, indicate the meaning of the values represented as a color scale.

P9, line 17: Please, replace “indicate” by “suggest”.

P9, figure 9: Please, indicate the meaning of the values at the axes.

P10, line 2: “The arrows in Figure 9C indicate”

Discussion

P10, line 30: Please, define “n” briefly here.

P10, line 38: “The adsorption mechanism was characterized in this study by using FTIR spectroscopy,…”

P11, line 5: “content”

P11, line 22: Please, remove “interior”.

P11, lines 27-28: Please, indicate in the figure the location of that “transparent circle” or remove this sentence. This observation anyway is very weak to support the affirmation of this sentence.

P11, line 27: Please, replace “indicating” by “suggesting”.

P11, line 30: Please, remove, “it was relatively easy for”.

P11, line 30: “Cs ions were able to enter into...”

P11, line 34: What’s the cytoderm? Please, find a more appropriate term and clarify if this sentence is referred to the active biosorption mechanism.

P11, line 49: Please, include another reference related to bacteria.

P12, line 4: “presence”.

P12, line 6: Please, replace “complicated” by “complex”.

P12, line 17: “The findings derived from this work provide…”

References

- Please, write scientific names in italics.

Author Response

Response to Reviewer 2 Comments

Point 1: The example provided, is it active or passive adsorption? Please, clarify this point.

Response 1: (page 2, line 7-9). The example provided is active adsorption. Active adsorption means accumulate intracellularly via energy-dependent transport. In the example, Cd2+ precipitated as a phosphate in the intracellular vacuoles instead of on the cell walls, indicating metal ions enter into cytoplasm via metabolism-dependent transport.

Point 2: Please, provide the brand of the light shaker.

Response 2: (page 2, line 50). We have added the brand of the light shaker according to the comment.

Point 3: Please, clarify whether standard deviations were calculated.

Response 3: (page 4, line 17-20). We have calculated the standard deviations and added the contents.

Point 4: Please, provide the software employed for calculations.

Response 4: (page 4, line 18). We have provided the software employed for calculations according to the comment.

Point 5: Please, include the brand of the BCA kit.

Response 5: (page 3, line 21). We have added the brand of the BCA kit according to the comment.

Point 6: The reference included seems to correspond to sugars only. Please, check it.

Response 6: (page 3, line 22). We have corrected the reference according to the comment.

Point 7: Please, provide more detail on how cells were prepared for FTIR and Raman.

Response 7: (page 3, line 40-43). We have provided the detail on sample preparation for FTIR and Raman according to the comment.

Point 8: Please, provide the brand of TEM.

Response 8: (page 4, line 10). We have supplemented the brand of TEM according to the comment.

Point 9: In this figure legend, please explain what the vertical bars are. Are they standard deviations? Please, clarify it.

Response 9: (page 4, line 28). Yes, the vertical bars in this figure are standard deviations. we have added this standard deviation, and Any p value less than 0.05 was considered significant.

Point 10: “concentrations”

Response 10: (page 5, line 5). We have corrected this mistake according to the comment.

Point 11: Please, replace “Total EPS was maximized” by “Total EPS production reached the highest level”.

Response 11: (page 5, line 5). We have modified this phrase according to the comment.

Point 12: In this figure legend, please explain what the vertical bars are. Are they standard deviations? Please, clarify it.

Response 12: (page 5, line 9). Yes, the vertical bars in this figure are standard deviations according to the comment.

Point 13: Please, provide suitable references for these equations.

Response 13: (page 5, line 16). We have added the references of these equations according to the comment.

Point 34: Please, explain the meaning of the numbers in brackets or remove them. The same for the equation presented in page 3.

Response 14: (page 6, line 3). The numbers in brackets have been removed according to the comment.

Point 15: It’s not necessary to denote the type of model employed within the figures as it is indicated in the figure legend.

Response 15: (page 6, line 19. page 7, line 1). We have corrected according to the comment.

Point 16: PO2- is repeated twice. Symmetric, asymmetric groups? Please, clarify this point.

Response 16: (page 7, line 10). We have removed one PO2-.

Point 17: The peak at 1659 corresponds to amide I rather than to amide II. Please, clarify this point.

Response 17: (page 7, line 13). We have corrected this mistake according to the comment.

Point 18: 1153 and 1516 peaks were not seen in Figure 5

Response 18: (page 7, line 20). We have replaced 1153 and 1516 peaks by 1154 and 1524 peaks according to the comment.

Point 19: To better follow the results, please represent a different figure for each FTIR and Raman spectra.

Response 39: (page 8, line 3-4). We have remade individual figure for FTIR and Raman spectra and the Raman spectra’s figure were added to supplementary materials

Point 20: Please, indicate the meaning of the values at the axes. Also, please remove the letter K following the element in the legend within the figure.

Response 20: The values in ordinate axes and abscissa represents the signal intensity and elemental characteristic X-ray wavelength, respectively. The elements in samples are determined by their wavelength, and the relative contents of the elements are determined by the signal intensity. We have removed the letter K in the figure according to the comment (page 9, line 1).

Point 21: “by product”

Response 21: (page 9, line 10). We have revised the mistake according to the comment.

Point 22: Please, replace “indicating” by “suggesting”.

Response 22: (page 9, line 13). We have replaced “indicating” by “suggesting” according to the comment.

Point 23: “Cs ions were loaded”. Please, remove “in or on Synechococcus PCC7002”.

Response 23: (page 9, line 15). We have corrected this sentence according to the comment.

Point 24: Please, replace “indicating” by “suggesting”.

Response 24: (page 9, line 16). We have replaced “indicating” with “suggesting” according to the comment.

Point 25: Please, indicate the meaning of the values represented as a color scale.

Response 25: (page 9, line 18). The values represented as a color scale means fluorescence intensity. EEM fluorescence spectra is comprised Excitation wavelength (y axis), Emission wavelength (x axis) and fluorescence intensity (z axis). The substance in samples are determined by their Excitation and Emission wavelength, and the relative contents of the substance are determined by the fluorescence intensity.

Point 26: Please, replace “indicate” by “suggest”.

Response 46: (page 10, line 12). We have replaced “indicate” with “suggest” according to the comment.

Point 27: Figure 9: Please, indicate the meaning of the values at the axes.

Response 27: (page 10, line 16). The values in ordinate axes and abscissa in Figure 9 represents the signal intensity and elemental characteristic X-ray wavelength, respectively. The elements in samples are determined by their wavelength, and the relative contents of the elements are determined by the signal intensity.

Point 28: “The arrows in Figure 9c indicate”

Response 28: (page 11, line 20). We have corrected this mistake according to the comment.

Point 29: Please, define “n” briefly here.

Response 29: n is Freundlich constant related to the adsorption intensity.

Point 30: “The adsorption mechanism was characterized in this study by using FTIR spectroscopy ,…”

Response 30: (page 11, line 23). We have corrected this sentence according to the comment.

Point 31: “content”

Response 31: (page 11, line 41). We have corrected this mistake according to the comment.

Point 32: Please, remove “interior”.

Response 32: We have removed “interior” according to the comment.

Point 33: Please, indicate in the figure the location of that “transparent circle” or remove this sentence. This observation anyway is very weak to support the affirmation of this sentence.

Response 33: We have removed this sentence according to the comment.

Point 34: Please, replace “indicating” by “suggesting”.

Response 34: The sentence has been removed. 

Point 35: Please, remove, “it was relatively easy for”.

Response 35: We have removed “it was relatively easy for” according to the comment.

Point 56: “Cs ions were able to enter into...”

Response 36: (page 12, line 12). We have corrected according to the comment.

Point 37: What’s the cytoderm? Please, find a more appropriate term and clarify if this sentence is referred to the active biosorption mechanism.

Response 37: (page 12, line 16). We have replaced “cytoderm” by “cytomembrane” according to the comment. This sentence is referred to the active biosorption mechanism and we revised this sentence for a better understanding of biosorption mechanism.

Point 38: Please, include another reference related to bacteria.

Response 38: (page 16, line 30). We have revised the reference according to the comment.q

Point 39: “presence”.

Response 39: (page 12, line 47). We have corrected according to the comment.

Point 40: Please, replace “complicated” by “complex”.

Response 60: (page 12, line 49). We have replaced “complicated” with “complex” according to the comment.

Point 41: “The findings derived from this work provide…”

Response 41: (page 13, line 8). We have modified this sentence according to the comment.

Point 42: Please, write scientific names in italics.

Response 42: These names were revised and modified according to the comment.